# Neural capacity limits on the responses to memory interference during working memory in young and old adults

**Jason Steffener** [1]*, **Daniel Barulli**[2], **Brianna Hill**[2]

**1** Interdisciplinary School of Health Sciences, University of Ottawa, Ottawa, Ontario, Canada, **2** Cognitive Neuroscience Division, Department of Neurology and Taub Institute for Research on Alzheimer's Disease and The Aging Brain, Columbia University College of Physicians and Surgeons, New York, New York, United States of America

* jason.steffener@uottawa.ca

**Data Availability Statement:** All participant level contrast maps, behavioral data and a link to the software used for administering this experiment are available at https://osf.io/xwajn/.

## Abstract

Advancing age affects the recruitment of task related neural resources thereby changing the efficiency, capacity and use of compensatory processes. With advancing age, brain activity may therefore increase within a region or be reorganized to utilize different brain regions. The different brain regions may be exclusive to old adults or accessible to young and old alike, but non-optimal. Interference during verbal working memory information retention recruits parahippocampal brain regions in young adults similar to brain activity recruited by old adults in the absence of external interference. The current work tests the hypothesis that old adults recruit neural resources to combat increases in age-related intrinsic noise that young adults recruit during high levels of interference during information retention. This experiment administered a verbal delayed item recognition task with low and high levels of an interfering addition task during information maintenance. Despite strong age-related behavioral effects, brain imaging results demonstrated no significant interaction effects between age group and the interference or memory tasks. Significant effects were only found for the interaction between interference level and memory load within the inferior frontal cortex, supplementary motor cortex and posterior supramarginal regions. Results demonstrate that neural resources were shared when facing increasing memory load and interference. The combined cognitive demands resulted in brain activity reaching a neural capacity limit which was similar for both age groups and which brain activation did not increase above. Despite significant behavioral differences the neural capacity limited the detection of age group differences in brain activity.

## Introduction

With advancing age, there comes a multitude of neural and cognitive changes. Current theories of cognitive aging suggest the existence of age-related declines in neural efficiency, neural capacity and neural compensation [1]. Efficiency describes the amount that brain activity

**Funding:** JS: #K01AG035061; National Institute on Aging; https://www.nia.nih.gov/ The funders had no role in study design, data collection and analysis, decision to publish, or preparation of the manuscript.

**Competing interests:** The authors have declared that no competing interests exist.

increases with increasing task demands. Capacity is the maximal level of activity that a brain region can reach. Compensation is a mechanism that allows individuals to successfully cope with increasing cognitive demands, external interference, or age-related neural structure decline [2].

The concept of compensation implies the need for something to be compensated for. This may generally be defined as advancing age; however, it implies that there are neural changes which require functional compensation. These changes include decreased functional neural efficiency and capacity within brain regions required for a task [3] or declining structural resources which affect brain function [4–6].

Functional brain mechanisms of compensation are described as upregulation, selection and reorganization of task related brain activity [1]. Neural resources which shift along a lifetime continuum [7] reflect upregulation or a decrease in neural efficiency [3]. The idea of selection is that compensatory resources used by old adults are also available to young adults; however, they are not optimal and not typically employed by the young adults. Reorganization of resources implies that the compensatory resources reflect the use of different and novel brain regions to complete a task.

Testing for the presence and nature of compensatory resources has often involved working memory tasks. This is due to the widely observed effects that age has on its associated behavioral and functional brain measures [8,9]. Previous work using letter items in a delayed item recognition (DIR) task found that the greatest age-group difference in brain activation was during the information maintenance period [10]. Follow-up investigations with the DIR task found that in addition to activation in prefrontal, precentral, insular and cingulate cortical regions, some, but not all, older adults showed brain activity within the parahippocampal gyrus which was associated with poorer performance. A follow-up study demonstrated that this additional activation was partially explained by reduced brain volume in the precentral cortex [4]. It was hypothesized that decreased integrity of the neural resources within the precentral gyrus required the older adults to alter their strategy, resulting in the observed parahippocampal brain activation. This speculation is supported by the report by Sakai et al. [11] that interference with information maintenance in young adults results in continual reactivation of the memory trace via activation within the parahippocampal gyrus [11]. Sakai et al. [12] also observed activation in the prefrontal cortex that was task related but did not differentiate based on levels of interference [12]. These results in young adults suggest that our previous finding in old adults may reflect a selection process of compensatory resources.

The current work tests whether the interaction between memory load and interference during information maintenance differs between young and old adults. The hypothesis is that older adults employ additional neural resources to combat increases in age-related intrinsic noise during performance of a working memory task. We stressed the available neural resources during a verbal DIR task by manipulating the number of items to remember and by introducing an interference numerical addition task during the information delay period. We predicted that increasing memory load would induce the use of additional neural resources within the prefrontal cortex and medial temporal lobe in old adults and would be comparable to the neural resources that young adults employ when faced with explicit interference with information maintenance [13].

## Materials and methods

### Participants

Informed consent, as approved by the Internal Review Board of the College of Physicians and Surgeons of Columbia University, was obtained in writing prior to study participation. Thirty-

nine healthy adults were scanned including 21 younger participants (5 men and 16 women mean (±s.d.) age = 25.33 (3.17); all right handed), and 18 healthy, older participants (10 men and 8 women; mean (± s.d.) age = 65.72 (4.53); all right handed). Participants were recruited from two other on-going studies at Columbia University within the Cognitive Neuroscience Division. Eligible participants from the other studies were asked whether they would be interested in participating in the current study. No matching took place between the two age groups, e.g. education, socio-economic status, etc. Future directions with this data will explore how lifetime exposures impact brain and cognitive measures in older adults. Therefore, matching between age groups was not undertaken.

Recruitment used market-mailing procedures for households within 10 miles of the northern Manhattan, NY, USA site. The aim was to equalize the recruitment approaches across the lifespan. Participants who responded to the mailing were telephone screened to ensure that they met basic inclusion criteria (right handed, English speaking, no psychiatric or neurological disorders, normal, or corrected-to-normal vision). All participants found eligible via the initial telephone screen were further screened in person with structured medical, neurological, psychiatric, and neuropsychological evaluations to ensure that they had no neurological or psychiatric disease or cognitive impairment. The screening procedure included a detailed interview that excluded individuals with a self-reported history of major or unstable medical illness, significant neurological history (e.g., epilepsy, brain tumor, stroke), history of head trauma with loss of consciousness for greater than 5 min or history of Axis I psychiatric disorder (American Psychiatric Association, 1994). Individuals taking psychotropic medications were also excluded. During the screening global cognitive functioning was also assessed with the Mattis Dementia Rating Scale and a score of at least 133 was required for admission to the study [14]. Participants were compensated for their participation in the study.

## Task

The experimental paradigm was a delayed item recognition task using letters as memory stimuli. During the information retention period, an interfering numeric addition task was introduced. The memory stimulus set was presented for 3 seconds, followed by a 0.5 second blank screen. The numeric stimuli were presented on the screen for 5 seconds and followed by a 0.5 second blank screen. A single letter probe stimulus was presented for at most 3 seconds; however, it was removed once the participant made a response and it was replaced with a blank screen. Following each trial, a blank screen was presented for an intertrial interval (ITI) of varying duration. The ITI length was the sum of the response time plus a variable additional time period. Thirty-two trials were presented in each scanning run and participants were engaged in three runs during scanning for a total of 96 trials. There were two memory loads and two levels of interference and 8 trials in each of these 4 conditions per scanning run. The task is shown in Fig 1.

## Letter recognition task

Lists of either two or six capital letters represented the low and high memory load conditions, respectively. Letters were drawn from the English alphabet and excluded the five vowels and Y to avoid word formation. Letters which were diagrammatically identical in their lower and uppercase form were also excluded. The exclusion list was: A, E, I, O, U, C, P, S, V, W, X, Z. Letters were presented using the `Courier` font presented with a size of 60. This is a serif font easing the distinction between letters such as lowercase "ell" (`l`), which is a possible probe letter, and the uppercase "eye" (`I`), which is not a possible letter in the study set. These letters are indistinguishable with san-serif fonts such as Arial (l, I). Even though such situations are

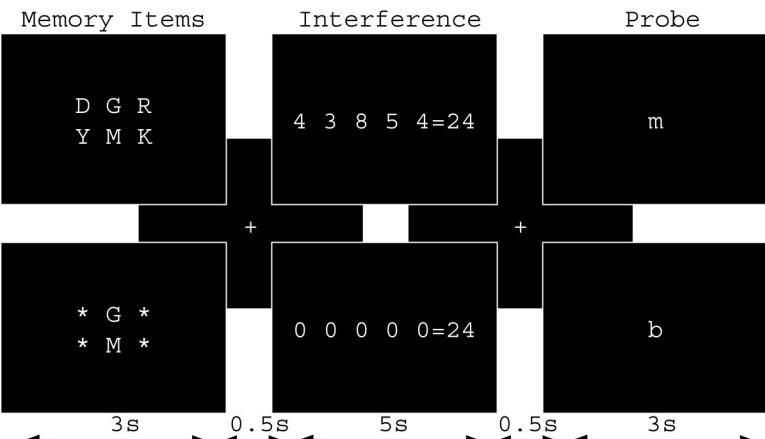

**Fig 1. Delayed item recognition task with interference.** One set of memory items were presented for three seconds and each trial consisted of either two or six letters. The memory set was removed from the screen and replaced by a crosshair for half a second. During the subsequent five seconds for information maintenance, the interference task was presented at two levels. Participants were required to determine if the five digits on the left hand side of the equal sign summed to equal the number on the right hand side. Low interference used five zeros and high interference used five non-zero digits. After presentation of another half second crosshair, a single lowercase probe letter was presented. Participants had to determine whether or not they recognized the letter as part of the set of memory items for the trial. Participants made timed yes/no button press responses to the interference task and to the memory task.

avoided due to the letter set chosen, this extra precaution was made. The probe was a single letter presented in the same font and size; however, in lowercase formatting. This minimizes visual matching between the study set and the probe and maximizes the use of auditory encoding of the memory set.

The ordering of the trials was carefully chosen to maximize design efficiency, as described below, and to minimize proactive interference. Therefore, a current trial's stimulus set could not include any of the letters from the previous trial's stimulus set or probe. The current trial's probe could likewise not match any of the letters in the previous trial's stimulus set or probe.

## Interference task

The interference during information retention required the participants to determine if the sum of five single digit numbers equaled a number presented opposite an equal sign, the answer. For low levels of interference, the five numbers to add were all zero and a positive probe (answer) was when the answer was also zero; a negative probe was when the answer was a number other than zero. For high levels of interference, the five numbers to add were all non-zero. Careful consideration was made when selecting the numbers to add and the answers. The summation of five single digits (excluding zero) drawn from a uniform distribution is a Gaussian distribution with a mean of 25. This makes extreme value answers, below 10 or above 40, highly unlikely. In order to minimize this bias and the predictability of the answers, the answers were chosen from a uniformly distributed set of numbers with a range of 5 to 45. These are the minimum and maximum values possible when adding five single digit numbers. Five randomly selected single digits that had this value as a sum were then used as the number list. The proportion of positive and negative probe trials was matched at fifty percent.

## Training

Training on the task had the intention of familiarizing the participants with the verbal information recognition task, the numeric addition task and the joint performance of the two tasks.

Successful training required evidence that participants understood the instructions and were able to perform above chance. The first training run presented 16 trials containing both letter loads with no interfering addition task along with complete instructions and accuracy feedback after every trial. The second training run presented 16 trials each with a memory load consisting of a single letter with the addition interference task along with accuracy feedback after every trial. The third training run presented 16 trials containing both memory letter loads and the addition interference task along with accuracy feedback for every trial. The final training run was identical to the actual task in the scanner, presenting 32 trials and no feedback. After every run, the researcher was provided performance metrics from which they could decide whether or not to repeat instructions or continue training. Successful completion of four training runs was required before participants entered the scanning room. All participants met training performance criteria for scanning. The task was written in MatLab by the author JS using the PsychToolbox (http://psychtoolbox.org) and is publicly available (https://github.com/steffejr/InterferenceLetterSternberg).

## Model efficiency

The design of this experiment used a trial based approach. Therefore, to maximize statistical efficiency, the trial order and intertrial interval times were carefully selected. The *a priori* analysis plan was to statistically model only the trials where the participant correctly determined whether or not the probe letter was part of the study set or not. Error trials were included as covariates of no interest in the model, collapsing across memory and interference levels.

Designing the experiment took into consideration the expected number of errors and the distribution of the response times when identifying the design with maximal efficiency. Behavioral pilot data was collected, results not shown, using a fixed intertrial interval in 20 younger and 20 older healthy adults. Simulated designs for the MRI phase of the experiment used the error rates and response times from this behavioral pilot. The contrasts of interest were calculated from this model and used to calculate the design matrix efficiency as:

$$efficiency = \frac{1}{trace(c^T \cdot (X^T X)^{-1} \cdot c)}$$

A total of one million simulations of random trial orders (low or high memory load) (1000) and intertrial interval distributions drawn from a Gamma distribution of times (1000) were tested. Each simulation used expected response times derived from the pilot data and an expected error rate. The efficiency of each of the estimated contrasts of interest were calculated from each of these simulations. The trial order and intertrial intervals of the three simulations providing the maximal efficiency across all contrasts of interest were retained and used for the MRI experiment.

## Behavioral analysis

A mixed level model tested for interactions between age group, memory load (2/6 letters) and interference (low/high addition) on accuracy and response time (RT). The intercept was considered a random effect while memory load and interference level were fixed effects. Model estimation used maximum likelihood and degrees of freedom were estimated using the Satterthwaite method [15]. Analyses were performed using Jamovi 1.0.7 [16–18].

## MRI data acquisition

MRI images were acquired in a 3.0 T Philips Achieva Magnet using a standard quadrature head coil. A T1-weighted scout image was acquired to determine the subject's position. One hundred and seventy contiguous 1 mm coronal T1-weighted images of the whole brain were acquired for each subject with an MPRAGE sequence using the following parameters: TR 6.6 ms, TE 3 ms; flip angle 8˚, acquisition matrix 256×256 and 240 mm field of view. Three functional scan sets were acquired, each of which included the collection of 240 functional images acquired using a gradient echo echo-planar imaging (EPI) sequence TE/TR = 20 ms/2000 ms; flip angle = 72˚; 112×112 matrix; in-plane voxel size = 2.0 mm×2.0 mm; slice thickness = 3.5 mm (no gap); 38 transverse slices per volume. Before the initiation of the task, four volumes were acquired and discarded to allow transverse magnetization immediately after radiofrequency excitation to approach its steady-state value. A neuroradiologist reviewed all T1 scans for potentially clinically significant findings, such as abnormal neural structure; no clinically significant findings were identified or removed. Geometric distortions in the EPI images were minimized using field map scans acquired using the same image dimensions and slice location as the functional scans, with TE = 2.6 and 4.3 ms and TR = 20 ms.

## Image pre-processing

All image pre-processing and statistical analyses used SPM8 (Wellcome Department of Cognitive Neurology). For each participant's EPI dataset, images were temporally shifted to correct for slice acquisition order using the first slice acquired in the TR as the reference. All EPI images were corrected for motion by realigning to the first volume of the first session. The T1-weighted (structural) image was coregistered to the first EPI volume using mutual information. This coregistered high-resolution image was used to determine the transformation into a standard space defined by the Montreal Neurological Institute (MNI) template brain supplied with SPM8. This transformation was applied to the EPI data and re-sliced using sinc-interpolation to 2 x 2 x 2 mm. Finally, all images were spatially smoothed with an 8 mm FWHM kernel.

## Participant level time-series analysis

The time series models crossed memory load (2 or 6 letters) and interference level (low or high) factors to create five regressors of interest. This includes regressors for the four cells of the ANOVA and a fifth regressor accounting for trials without responses or incorrect responses regardless of load or interference level. Each trial was modeled as a rectangular epoch lasting from the start of the trial until the trial specific response was made (i.e., the three second stimulus plus the six second retention period plus the RT from the start of the probe presentation) [19]. All regressors of the time series models were convolved with a standard double-Gamma model of the hemodynamic response function [20].

## Group analyses

Brain imaging group analysis used a 2x2x2 ANOVA crossing age group, memory load and interference load. Each participant provided four contrasts, one for each cell of the ANOVA, to the group analyses collapsing across the two scanning runs of the task.

# Results

## Summary of behavioral results

There were significant interactions between age group and memory load for memory accuracy and between age group and interference for memory response time. All main effects for

accuracy and response time were also significant. On the interference task, there was a significant interaction between age group and interference for accuracy and significant main effects for age group and interference for response time. Details for these results are reported as follows with unstandardized effect sizes. There is currently a lack of consensus for calculating standardized effect sizes based on the manner in which mixed models partition variance [21]. Therefore, current recommendations are followed [22] and unstandardized effect sizes are reported in their original units of seconds or percent accuracy.

### Task accuracy for memory

A mixed level model fit by restricted maximum likelihood (REML) tested for interactions between memory load (2 or 6 letters), interference level (low or high) and age group (young or old). The intercept (participant ID) was considered a random effect while memory load, interference and age group were fixed effects. There was a significant interaction between age group and memory load ($F(1, 2999) = 10.10$, $p = 0.002$, unstandardized effect size (uES) = -0.071 percent). Post-hoc tests with Bonferroni correction for multiple comparisons show that this interaction was driven by worse performance for 6 versus 2 letters of memory load in both age groups (Young: Diff = 0.091, Z = 6.03, $p_{Bonf} < 0.001$; Old: Diff = 0.162, Z = 9.83, $p_{Bonf} < 0.001$), worse performance at 6 letters for the old group compared to the young (Diff = 0.107, Z = 4.44, $p_{Bonf} < 0.001$) and worse performance in the old group at six letters versus the young at 2 letters (Diff = 0.199, Z = 8.21, $p_{Bonf} < 0.001$). All other interactions were non-significant (age group by memory load by interference level: $F(1, 2998) = 3.64$, $p = 0.056$, uES = 0.085; age group by interference level: $F(1, 3017) = 0.23$, $p = 0.632$, uES = -0.011; memory load by interference level: $F(1, 2998) < 0.001$, $p = 0.99$, uES = -0.00026). All three main effects were significant; age group: $F(1, 36.8) = 11.24$, $p = 0.002$, uES = -0.072, worse performance by old age group (Diff = 0.072, Z = 3.35, $p_{Bonf} < 0.001$); memory load: $F(1, 2999) = 128.20$, $p < 0.001$, uES = -0.13, worse performance at 6 letters (Diff = 0.13, Z = 11.30, $p_{Bonf} < 0.001$); interference level: $F(1, 3017) = 49.10$, $p < 0.001$, uES = -0.079, worse performance at high interference (Diff = 0.079, Z = 7.01, $p_{Bonf} < 0.001$). The intraclass correlation coefficient for the random intercept was 0.033 (intercept variance = 0.0032, residual variance = 0.094). The Akaike Information Criteria (AIC) for model fitness was 1533. Marginal means with standard errors are shown in Fig 2A.

### Response time for memory

Using the same model as described above, there were significant interactions between memory load and interference level ($F(1, 2996) = 13.82$, $p < 0.001$, uES = -0.12 seconds) and age group and interference level ($F(1, 2996.1) = 5.40$, $p = 0.020$, uES = 0.077). Post-hoc tests with Bonferroni correction for multiple comparisons show that this interaction was driven by significantly longer response times for 6 letters compared to 2 at low interference (Diff = 0.39, Z = 17.29, $p_{Bonf} < 0.001$) and high interference (Diff = 0.27, Z = 11.28, $p_{Bonf} < 0.001$), high versus low interference at 2 letters (Diff = 0.31, Z = 13.28, $p_{Bonf} < 0.001$) and 6 letters (Diff = 0.19, Z = 7.99, $p_{Bonf} < 0.001$) and high interference at 6 letters versus low interference at 2 letters (Diff = 0.58, Z = 24.73, $p_{Bonf} < 0.001$) and low interference at 6 letters versus high interference at 2 letters (Diff = 0.083, Z = 3.54, $p_{Bonf} = 0.002$). The remaining interactions were non-significant (age group by memory load by interference level ($F(1, 2996) = 1.26$, $p = 0.26$, uES = -0.074); age group by memory load ($F(1, 2996) = 0.48$, $p = 0.49$, uES = 0.023)). All three main effects were significant, memory load: $F(1, 2996) = 403.20$, $p < 0.001$, uES = 0.33, longer at 6 letters (Diff = 0.33, Z = 20.1, $p_{Bonf} < 0.001$), interference: $F(1, 3000) = 224.6$, $p < 0.001$, uES = 0.25, longer at high interference (Diff = 0.25, Z = 15.00, $p_{Bonf} < 0.001$) and age group:

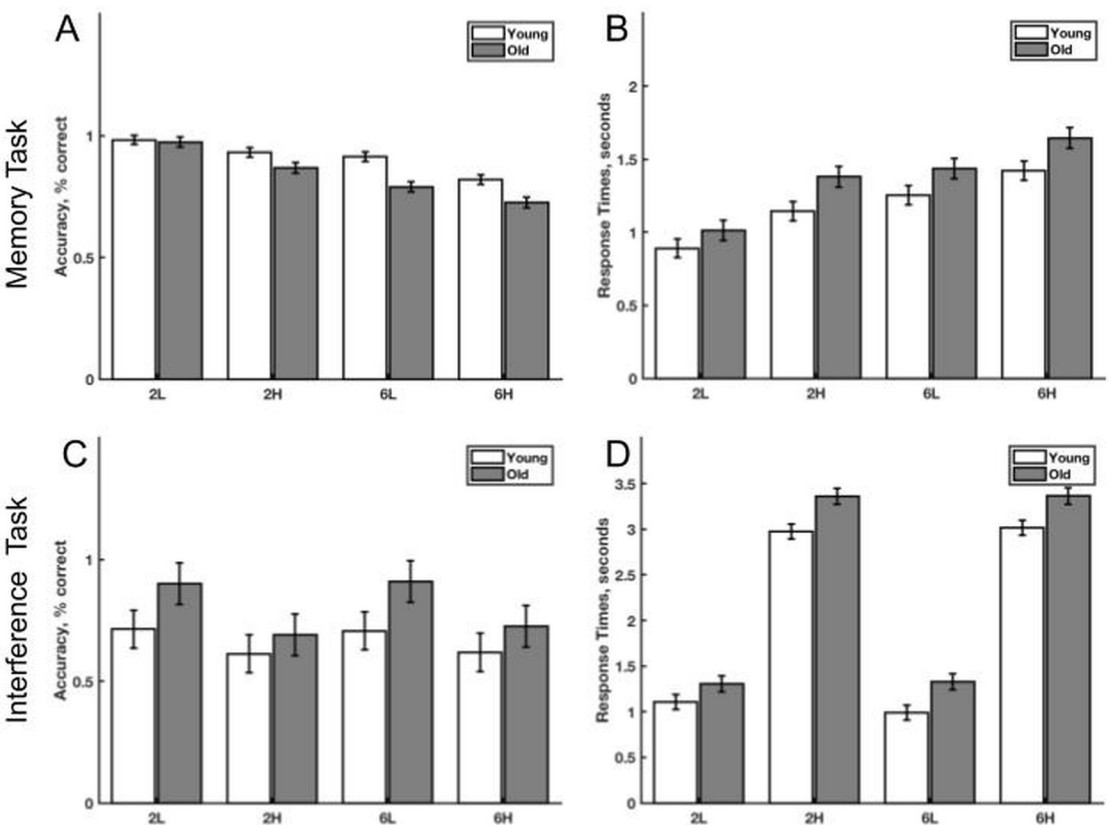

**Fig 2. Behavioral results.** Behavioral results for memory task and interference task. A) Marginal mean and standard errors for accuracy on the memory task expressed as percent correct, B) Marginal mean and standard errors for response time on the memory task expressed in seconds, C) Marginal mean and standard errors for accuracy on the interference task, D) Marginal mean and standard errors for response times on the interference task. 2L: Two letter memory load, low level interference task, 2H: Two letter memory load, high level interference task; 6L: Six letter memory load, low level interference task; 6H: Six letter memory load, high level interference task.

$F(1, 36.8) = 4.38$, $p = 0.043$, uES = 0.19), longer for older adults (Diff = 0.19, $Z = 2.09$, $p_{Bonf} = 0.036$). The intraclass correlation coefficient for the random intercept was 0.28 (intercept variance = 0.079, residual variance = 0.20). The Akaike Information Criteria (AIC) for model fitness was 3982. Marginal means are shown in Fig 2B with standard errors.

### Task accuracy for interference task

There was a significant interaction between age group and interference level ($F(1, 3000) = 31.25$, $p < 0.001$, uES = -0.10 percent). Post-hoc tests with Bonferroni correction for multiple comparisons show that this interaction was driven by significantly worse performance at high versus low interference levels in the young (Diff = 0.09, $Z = 7.63$, $p_{Bonf} < 0.001$) and old (Diff = 0.20, $Z = 14.48$, $p_{Bonf} < 0.001$) groups and a main effect of interference level ($F(1, 3000) = 251.10$, $p < 0.001$, uES = -0.15) driven by lower performance at high interference (Diff = 0.15, $Z = 15.80$, $p_{Bonf} < 0.001$). All other interactions and main effects were not significant; the three way effect ($F(1, 3000) = 0.18$, $p = 0.67$, uES = 0.015), memory load by interference level ($F(1, 3000) = 1.146$, $p = 0.28$, uES = 0.020), age group by memory load ($F(1, 3000) = 1.40$, $p = 0.24$, uES = 0.022), main effect of age group ($F(1, 37) = 1.59$, $p = 0.21$, uES = 0.14), memory load ($F(1, 3000) = 1.37$, $p = 0.24$, uES = 0.011). The intraclass correlation coefficient for the

random intercept was 0.667 (intercept variance = 0.13, residual variance = 0.063). The Akaike Information Criteria (AIC) for model fitness was 461.1. Marginal means are shown in Fig 2C with standard errors.

## Response time for interference task

There were significant main effects of age group ($F(1, 36.4) = 9.14$, $p = 0.005$, uES = 0.34 seconds) and interference level ($F(1, 3001) = 7144$, $p < 0.001$, uES = 2.02). Post-hoc tests with Bonferroni correction for multiple comparisons show that the old group performed slower than the young group (Diff = 0.34, $Z = 3.02$, $p_{Bonf} = 0.003$). High versus low levels of interference resulted in slower response times (Diff = 2.02, $Z = 84.50$, $p_{Bonf} < 0.001$). The three-way interaction was not significant ($F(1, 3000) = 0.90$, $p = 0.34$, uES = -0.090) as were all two-way interactions; age group and memory load ($F(1, 2996) = 0.016$, $p = 0.90$, uES = 0.0060), age group and interference level ($F(1, 3000) = 1.25$, $p = 0.26$, uES = 0.053), memory load by interference level ($F(1, 3000) = 0.30$, $p = 0.58$, uES = 0.026). The intraclass correlation coefficient for the random intercept was 0.22 (intercept variance = 0.12, residual variance = 0.42). The Akaike Information Criteria (AIC) for model fitness was 6166. Marginal means are shown in Fig 2D with standard errors.

## Speed accuracy tradeoff (IES)

To explore the finding of better, but slower, responses on the interference task by the old adults the inverse efficiency score was calculated by dividing mean accuracy by median response times [23]. This is a method of testing for speed-accuracy tradeoffs. A two-by-two repeated measures ANOVA demonstrated a significant three-way interaction between age group, interference level and memory load ($F(1, 37) = 4.47$, $p = 0.041$, $\eta^2_p = 0.11$). There was also a main effect of interference level ($F(1, 37) = 466.2$, $p < 0.001$, $\eta^2_p = 0.93$). All two way interactions were non-significant; memory load and interference level ($F(1, 37) = 4.08$, $p = 0.051$, $\eta^2_p = 0.10$), age group and interference level ($F(1, 37) = 0.74$, $p = 0.40$, $\eta^2_p = 0.02$) age group and memory load ($F(1, 37) = 1.99$, $p = 0.17$, $\eta^2_p = 0.051$). The main effect of memory load was non-significant ($F(1,37) = 3.22$, $p = 0.07$, $\eta^2_p = 0.08$). Marginal means for the three way interaction demonstrate that at high levels of interference, when memory loads increase the IES of the older adults significantly declines (mean difference = -0.35, $t = 3.54$, $p_{Bonf} = 0.02$), while in the young adults it does not significantly change (mean difference = -0.014, $t = 0.15$, $p_{Bonf} = 1.00$).

## Brain results

Brain imaging results corrected for multiple comparisons with an activation height threshold of $p < 0.001$ and cluster extent threshold of $p < 0.05$. This extent threshold was achieved using a size limit of $k = 153$ to minimize false positive findings to an alpha of 0.05. These thresholds were calculated using the updated AFNI tool 3dClusterSim and 10,000 Monte Carlo simulations with a non-Gaussian spatial autocorrelation function (ACF) determined based on individual residual maps averaged across all participants with bi-sided thresholding [24,25]. This approach uses updated methods based on findings related to accurate correction for multiple comparisons [26].

The two-way interaction between memory load and interference level had three significantly large clusters of activity within the left inferior frontal gyrus, midline supplementary motor cortex and extending from the right posterior supramarginal into white matter. These results along with post-hoc tests investigating the interaction effect using a Bonferroni corrected alpha < 0.001 are shown in Fig 3A and Table 1. Fig 4 shows bar plots of the activation within the cluster maxima for these three clusters.

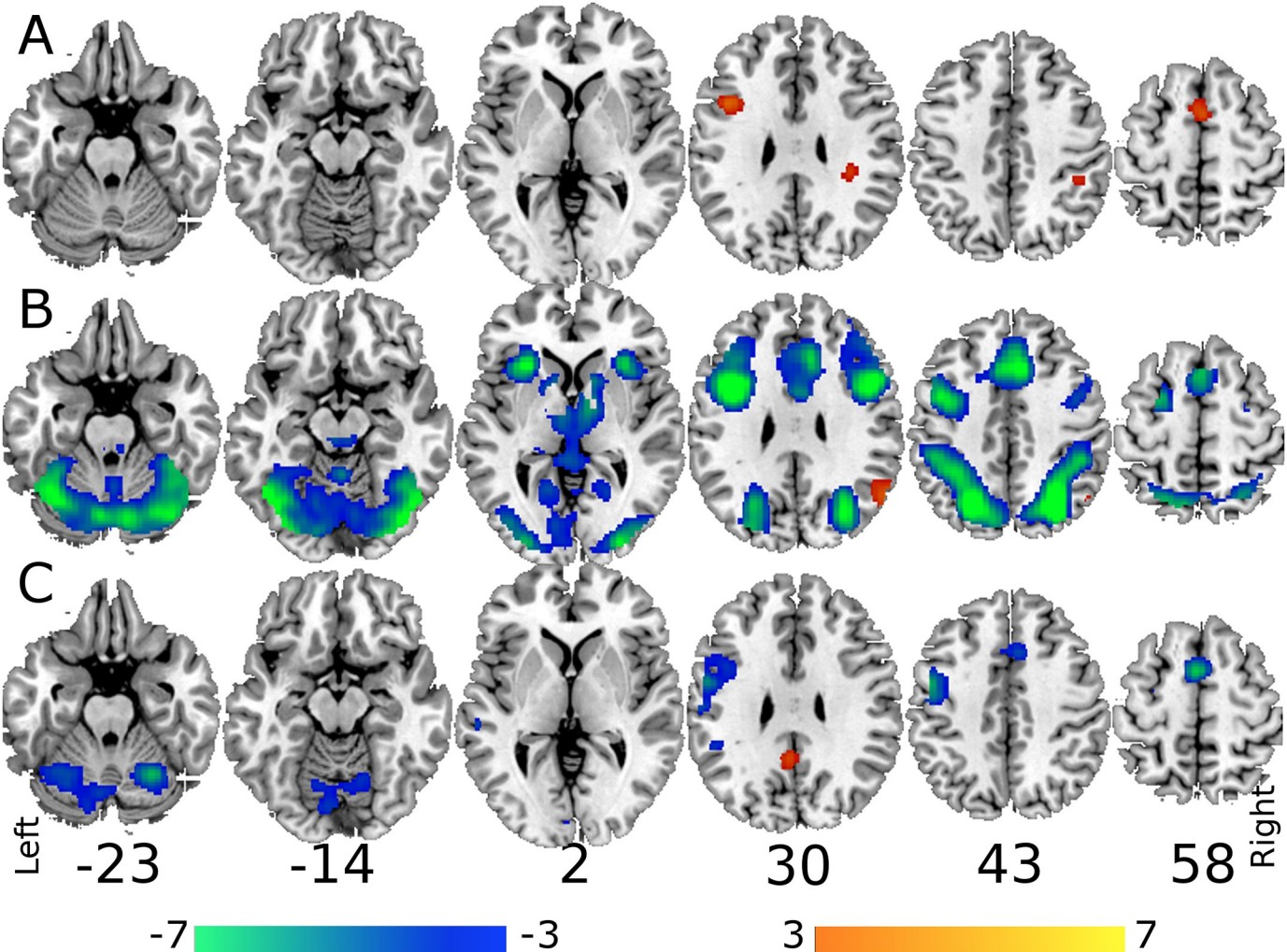

**Fig 3. Brain imaging results.** Overlay of results for A) Contrast testing the two-way interaction between memory load and level of interference, 6 letters at low interference plus 2 letters at high interference greater than 6 letters at high interference plus 2 letters at low interference; B) Memory load, cool colors represent 6 > 2 letters and warm colors represent 2 > 6 letters; and C) Interference level, cool colors represent High > Low levels of interference and warm colors represent Low > High levels of interference. The height threshold for all results was p < 0.001 and an extent of k > 153 contiguous voxels.

**Table 1. Interaction of memory load and interference brain imaging results.**

| Region | Lat. | x | y | z | t | k | Post-Hoc Tests | | | | | |
|---|---|---|---|---|---|---|---|---|---|---|---|---|
| | | | | | | | 2H-2L | 6H-2H | 6H-2L | 6L-2L | 6L-2H | 6H-6L |
| Inferior frontal g. | L | -40 | 10 | 28 | 4.81 | 222 | ** | | ** | ** | ** | |
| - - | | 0 | 4 | 60 | 4.49 | 181 | ** | | ** | ** | | |
| Supp MA | R | 4 | 16 | 52 | 3.63 | - - | ** | ** | ** | ** | ** | |
| - - | R | 34 | -34 | 32 | 4.13 | 177 | | | | | | |
| Post-Supra | R | 44 | -38 | 42 | 3.63 | - - | | | ** | ** | ** | |
| - - | R | 34 | -24 | 36 | 3.41 | - - | | | | | | |

Lat: Laterality, mid: midline, k: cluster size, - -: a local maxima within a larger cluster. Supp MA: Supplementary motor cortex. Post-Supra: Posterior supramarginal cortex. Post-hoc tests performed using Bonferroni corrected alpha = 0.001. (T > 3.8635)

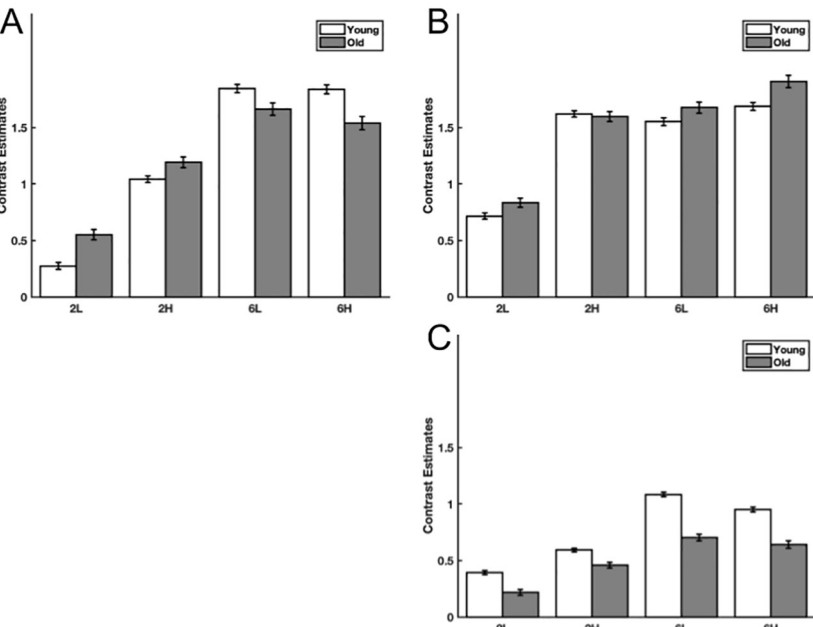

**Fig 4. Bar plots of effects in cluster maxima.** Bar plots of group mean levels with 95% confidence intervals of task related signal change within the cluster maxima for each task condition. A) Left inferior frontal gyrus, B) Supplementary Motor Cortex C) Posterior supramarginal cortex. 2L: Two letter memory load, low level interference task, 2H: Two letter memory load, high level interference task; 6L: Six letter memory load, low level interference task; 6H: Six letter memory load, high level interference task.

Main effects of memory load showed extensive activation throughout the brain including bilateral inferior parietal into inferior and superior occipital regions, right precentral extending through the cingulate into the left supplementary motor area, left inferior prefrontal areas extending back to the precentral gyrus and the left insula, see Fig 3B and Table 2. Main effects of interference had activation in the right precentral gyrus extending to the supramarginal gyrus, bilateral supplementary motor areas, bilateral cerebellum and right middle and superior

**Table 2. Main effect of memory load brain imaging results.**

| Region | Lat. | x | y | z | t | k |
|---|---|---|---|---|---|---|
| Positive Direction | | | | | | |
| Angular gyrus | L | 56 | -60 | 34 | 5.14 | 267 |
| Negative Direction | | | | | | |
| Inferior Parietal | L | 28 | -70 | 42 | 9.32 | 25740 |
| Inferior Occipital | L | 42 | -76 | -12 | 8.45 | - - |
| Superior Occipital | R | -24 | -64 | 44 | 8.41 | - - |
| Precentral | R | -42 | 6 | 32 | 8.83 | 7857 |
| Supplementary Motor Area | L | 2 | 16 | 52 | 8.09 | - - |
| Cingulum | L | 8 | 20 | 40 | 7.59 | - - |
| Inferior Frontal Operculum | L | 50 | 10 | 30 | 7.74 | 2499 |
| Precentral | L | 38 | -2 | 52 | 6.53 | - - |
| Inferior Frontal | L | 50 | 30 | 26 | 4.74 | - - |
| Insula | L | 32 | 22 | 4 | 6.48 | 587 |

Lat: Laterality, mid: midline, k: cluster size, - -: a local maxima within a larger cluster. Thresholds were alpha < 0.05 and cluster extent of 153.

**Table 3. Main effect of interference brain imaging results.**

| Region | Lat. | x | y | z | Z | k |
|---|---|---|---|---|---|---|
| Positive Direction | | | | | | |
| Precuneus | R | -6 | -54 | 18 | 4.67 | 23 |
| Negative Direction | | | | | | |
| Precentral | R | -52 | -4 | 38 | 6.45 | 1036 |
| Precentral | R | -44 | -6 | 46 | 5.72 | - - |
| SupraMarginal | R | -58 | -18 | 28 | 4.25 | - - |
| Supplementary MotorArea | R | -2 | 4 | 60 | 6.3 | 646 |
| Supplementary MotorArea | L | 4 | 14 | 50 | 4.7 | - - |
| Cerebellum, 6 | L | 28 | -60 | -22 | 5.9 | 1580 |
| Cerebellum, 6 | R | -28 | -62 | -20 | 4.23 | - - |
| Cerebellum, 6 | R | -6 | -72 | -20 | 4.2 | - - |
| Superior Temporal | R | -50 | -40 | 24 | 4.79 | 360 |
| Superior Temporal | R | -58 | -30 | 4 | 4.08 | - - |
| Middle Temporal | R | -48 | -36 | 6 | 4.05 | - - |

Lat: Laterality, mid: midline, k: cluster size, - -: a local maxima within a larger cluster. Thresholds were alpha < 0.05 and cluster extent of 153.

gyri of the temporal cortex, see Fig 3C and Table 3. The three-way interaction demonstrated no significant results in either direction. Both two-way interactions involving age group and the main effect of age demonstrated no significant results.

## Discussion

Age, memory load and level of interference from an addition task during information retention were all found to influence task accuracy and response times; however, their effects were not completely additive for performance on the memory task. The level of interference impacted both accuracy and response time on the interfering addition task with a significant interaction for accuracy. Although older adults were significantly more accurate on the addition task, they were also slower, as compared to the younger group. Analysis of the inverse efficiency score supports the hypothesis that the old adults are making a speed accuracy tradeoff when completing the interference task. Results from the brain imaging data suggest that memory load and interference pushed neural activation levels up to their neural capacity limits for both age groups.

This study had robust task-related memory load and interference level brain activation findings. There were no significant age-group differences, nor interactions between age groups and the other factors at the conservative cluster extent threshold used for correction of multiple comparisons. Significant effects were found in the interaction between interference and memory load within the inferior frontal gyrus, supplementary motor cortex and the posterior supramarginal cortex. For both age groups there were effects of interference at memory loads of two letters; however, there were no interference effects at memory loads of six letters. This finding was similar in all identified brain regions, with overall reductions within the posterior supramarginal cortex cluster. These results suggest that the demands of this task were sufficiently high such that neural capacity may have been reached [2,3] at six letters and low interference levels for both age groups. Operating at neural capacity therefore limits one's ability to recruit additional within region neural resources to respond to increasing cognitive demands [27]. The result is the observed lack of age group interaction effects.

This interpretation suggests that increases in memory load and increases in interference both place greater demands on the neural resources within the same regions. This is supported by the fact that the increases in brain activity from the main effect of interference are all within regions also identified as having greater brain activity with the main effect of memory load. The only exception is a cluster in the superior temporal gyrus. These brain regions are also inclusive of those identified as having memory load, but not age effects in a similar study [28]. Neural capacity limitations may hamper the recruitment of compensatory increases of task related brain activity thereby limiting the ability to detect the expected interaction effects of this study.

This study was designed around the hypothesis that older adults would employ information rehearsal mechanisms similar to those previously shown in the parahippocampus or the pre-central gyrus [4,10,11]. Results did not demonstrate significant findings in these regions. It is plausible that the intrinsic noise hypothesis of the current study is not appropriate. Observed behavioral differences may reflect age-related declines in the availability of attention [29]. Meanwhile, the neural capacity limits bolster the idea of shared neural resources for working memory operations that draw from the same capacity limited system [30,31]. Therefore, cognitive capacity limitations of the working memory system may reflect a neural capacity which limits the engagement of compensatory neural resources resulting in the observed lower task performance levels.

The interruption of information retention during working memory may also be interpreted as a study of dual-task effects [31–34]. One observation from dual-task studies is the sharing of neural resources across concurrent tasks, specifically within lateral prefrontal cortical (LPFC) regions [32,33]. With this in mind, the current brain imaging results are in line with other findings from dual-task paradigms demonstrating increased activation within inferior frontal and parietal cortices [35]. Age effects in dual-task studies appear to be limited such that the neural substrates utilized when managing dual-tasks are observed as largely similar across age groups [36,37].

The finding of neural capacity limitations within the fMRI signal warrants further exploration. The dynamic range of the fMRI BOLD signal is shown to be related to task performance and successful cognitive aging [38] and is adaptive [39]. These authors discuss capacity within an adaptive model developed through evolutionary pressure and influenced by participation in cognitively engaged physical activities, i.e., foraging. The notion that neural capacity is plastic is supported by findings that the relationships between brain activation and cognitive demands are moderated by cardiorespiratory fitness levels [40]. Future work will explore the relationships between assessments of cognitive capacity and neural capacity and the role of individual differences in lifestyles and behaviors.

There are limitations in this work which also need to be addressed. It is possible that despite attempts at designing the task with minimal demands for the low level of interference, there was enough interference to interrupt the information rehearsal process. Future studies should employ additional control conditions that present an uninterrupted maintenance period, i.e., visual stimuli with no cognitive processing. The current analyses also collapsed across all phases of the task, information encoding, maintenance and recognition. This is a strength and a limitation. This approach increased the power of the statistical model [41] by avoiding collinearity [42] and minimized the number of covariates in the model facilitating interpretation. However, it also decreased the ability to detect age-related differences in individual phases of the task. Future directions will explore the individual task phases to identify how they differ in their relationship with task performance and how they are affected by interference.

The current work used two levels of memory load: 2 and 6 letters. Previous work with a similar task has used 5 or 6 letters as the maximal load level [10,11]. At this memory load both

young and old adults had some level of decrease in accuracy. The intent of the current work was to implement a task that was demanding; however, not too demanding as to exceed cognitive capacity limits. The addition of a 2 letter memory load condition allows for testing of memory load effects. The use of 2 letters was considered low enough to not impact accuracy; however, high enough to keep participants engaged with the task. An ideal experimental design would use additional memory load levels to further explore how behavior and brain activity dynamically change as cognitive demands increase [43,44].

This study demonstrated that despite significant behavioral differences during performance of a memory task, there were no significant age group differences in brain activity. The increases in interference and in memory load shared neural resources to push brain activity up to capacity limits. Therefore, there may have been no additional available resources for either age group to recruit and respond to increases in cognitive demand.

## Author Contributions

**Conceptualization:** Jason Steffener.

**Data curation:** Daniel Barulli, Brianna Hill.

**Formal analysis:** Jason Steffener.

**Funding acquisition:** Jason Steffener.

**Investigation:** Jason Steffener.

**Methodology:** Jason Steffener.

**Project administration:** Daniel Barulli, Brianna Hill.

**Software:** Jason Steffener.

**Writing – original draft:** Jason Steffener.

**Writing – review & editing:** Jason Steffener.

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
