## [Decision Letter · Decision Letter 0]

16 Jun 2020

PONE-D-20-11960

Neural capacity limits on the responses to memory interference during working memory in young and old adults

PLOS ONE

Dear Dr. Steffener,

Thank you for submitting your manuscript to PLOS ONE. After careful consideration by 2 Reviewers and an Academic Editor, all of the critiques of both Reviewers, especially the statistical concerns of Reviewer #2, must be addressed in detail in a revision to determine publication status. If you are prepared to undertake the work required, I would be pleased to reconsider my decision, but revision of the original submission without directly addressing the critiques of the two Reviewers does not guarantee acceptance for publication in PLOS ONE. If the authors do not feel that the queries can be addressed, please consider submitting to another publication medium. A revised submission will be sent out for re-review. The authors are urged to have the manuscript given a hard copyedit for syntax and grammar.

We look forward to receiving your revised manuscript.

Kind regards,

Stephen D. Ginsberg, Ph.D.

Section Editor

PLOS ONE

**Comments to the Author**

1. Is the manuscript technically sound, and do the data support the conclusions?

Reviewer #1: Yes

Reviewer #2: No

2. Has the statistical analysis been performed appropriately and rigorously? 

Reviewer #1: Yes

Reviewer #2: No

3. Have the authors made all data underlying the findings in their manuscript fully available?

Reviewer #1: No

Reviewer #2: No

4. Is the manuscript presented in an intelligible fashion and written in standard English?

Reviewer #1: Yes

Reviewer #2: Yes

5. Review Comments to the Author

Reviewer #1: Review: Neural capacity limits on the responses to memory interference during working memory in young and old adults. (PONE-D-20-11960).

Summary

The authors present a study in which they examine the differential effects of memory load and interference on neural activity in younger and older adults. The major finding was an unexpected one, showing that the neural capacity limit was similar for both age groups, even though behavioural results were found to be better in younger adults. The research design is sound, and the findings are interpreted appropriately based on the data that was collected. At this stage, my main aim is to offer suggestions for revisions that would further increase the clarity and generalizability of the results. I offer these suggestions below, in the order in which they appear in the paper.

• The second sentence of the paper (line 44-48) is lengthy, and the internal syntax of the sentence is difficult to read. I would suggest re-writing may help.

• Line 78-80 – The authors describe the hypothesis that ‘older adults employ additional neural resources’ in a working memory task. I do not contest that this is true, but it would be good at this point to set out which neural resources these are suggested to be. This is especially true as this becomes important later on in the paper during data analysis.

• Line 88-91 – Were participants matched for things like intelligence, education, etc? If so, authors should describe this matching. If not, they should explain why matching was not necessary for this investigation.

• Line 103-105 –Authors should described if any of the participants fall below 133 on the Mattis Dementia Rating Scale? If so, how many?

• Line 134 (and the whole design) – How did the authors come to decide on 2 and 6 letters as the 2 levels of memory load? In older studies of working memory, a capacity of 7 was established (Miller, 1956), while more recent work on strictly visual working memory has found the limit is more like 4 (Cowan, 2001). In either case, these load levels are either both under the capacity, or one is above and the other is below capacity, which makes interpretation of these results challenging. Additionally, memory load has been shown on a continuous scale both behaviourally and neurophysiologically (e.g. Vogel & Machizawa, 2004, Figure 3), so it may have been better to use some intermediate levels of load here as well. I am not asking for further data collection here, but the authors should include a discussion of why they made these decisions, as well as how the results may have differed based on different decisions having been made.

• Line 150-159 – The authors should clarify the details of the ‘math’ in this task. Upon first reading, it seemed that answers and digits were chosen randomly from distributions (as described), which would have meant that almost all of the examples given to participants would have been ‘incorrect’. Having looked at this again, I no longer think this was true, but I also am not sure about this. The authors should clearly set out the proportion of trials that were ‘correct’ and ‘incorrect’, and then describe the specific way in which the numbers were acquired from said distributions.

• Throughout the results, it would be good to see an estimate of effect size along with the other statistical information that is provided.

• It is also now very close to standard that data (and analysis code, where available) are uploaded to a repository such as OSF at the time of review. I appreciate that the corresponding author is willing to share the data if requested, but increasing transparency further by uploading it publicly would be better.

• Line 249-250 – The description of the age group x memory load interaction seems to be missing its p value.

• Line 376-378 – If the older adults are showing signs of speed/accuracy trade-off, it would be good for the authors to examine this by looking at something like Inverse Efficiency Scores for both younger and older adults.

• Line 386-390 – This is where having used more/intermediate levels of memory load in the experimental design would have been helpful in having more diverse results to examine.

• Line 391-392 – The phrasing in this sentence is strange to me. I believe I know what the authors intended to say, but to say that ‘increasing memory load and interference share neural resources’ is a bit opaque. Maybe better to say that memory and interference share resources first, and then describe increases/decreases in each

• Line 411-412 – This statement could be expanded on to further explain what the general findings are in terms of dual-task effects.

• Line 425-426 – This is an important thing to consider, and while I appreciate the reasons for not analysing things at each stage of the design, I think it may be a useful analysis to include if it is possible to do at this point. In a related study that we did (Wilbiks & Dyson, 2016), we found that individual differences at different phases of the perception/memory process were predictive of capacity for audiovisual integration. A similar finding would be really interesting here!

Review Signed: Jonathan Wilbiks, 8 June 2020.

Reviewer #2: The present study compared differences between younger and older adults in performance of verbal working memory with interference tasks during the delay period. The authors found that the two groups had significant behavioral difference but not much difference in brain activation. They concluded that such null effects in brain activation were due to compensation mechanisms for the older adults. Although the results are generally reasonable, the interpretations and the statistical methods need revision.

1) The results here were not adequately presented. For example, the behavioral results did not indicate those significant effects. What’s the p value? What’s the effect size?

2) Imaging results were not presented with a standard form. For example,”Brain imaging results were first interpreted using a correction for multiple comparisons with an activation height threshold of p < 0.001 and cluster extent threshold of p < 0.05. ”

3) References were carelessly inserted. For example, the first reference they used was “Author correction”instead of normal article. “These changes include decreased functional neural efficiency and capacity within brain regions required for a task, [3] or declining”[3] was in the wrong place.

4) The interpretation of the results could be alternative. The tasks are simple especially for the load 2 condition. There could be floor effect for the two groups. Imaging results are further less sensitive than the behavioral results, so as not to differentiate between the two groups.

5) The degree-of-freedom in the statistical results is weird for like thousands.

6. PLOS authors have the option to publish the peer review history of their article (what does this mean?). If published, this will include your full peer review and any attached files.

**Do you want your identity to be public for this peer review?** For information about this choice, including consent withdrawal, please see our Privacy Policy.

Reviewer #1: Yes: Jonathan M. P. Wilbiks

Reviewer #2: No

---

## [Author Response · Author response to Decision Letter 0]

26 Jun 2020

Reviewer #1: Review: Neural capacity limits on the responses to memory interference during working memory in young and old adults. (PONE-D-20-11960).

Summary

The authors present a study in which they examine the differential effects of memory load and interference on neural activity in younger and older adults. The major finding was an unexpected one, showing that the neural capacity limit was similar for both age groups, even though behavioural results were found to be better in younger adults. The research design is sound, and the findings are interpreted appropriately based on the data that was collected. At this stage, my main aim is to offer suggestions for revisions that would further increase the clarity and generalizability of the results. I offer these suggestions below, in the order in which they appear in the paper.

• The second sentence of the paper (line 44-48) is lengthy, and the internal syntax of the sentence is difficult to read. I would suggest re-writing may help.

 The first paragraph has been rewritten as follows:

With advancing age, there comes a multitude of neural and cognitive changes. Current theories of cognitive aging suggest the existence of age-related declines in neural efficiency, neural capacity and neural compensation (Cabeza et al., 2018). Efficiency describes the amount that brain activity increases with increasing task demands. Capacity is the maximal level of activity that a brain region can reach. Compensation is a mechanism that allows individuals to successfully cope with increasing cognitive demands, external interference, or age-related neural structure decline (Barulli & Stern, 2013). 

• Line 78-80 – The authors describe the hypothesis that ‘older adults employ additional neural resources’ in a working memory task. I do not contest that this is true, but it would be good at this point to set out which neural resources these are suggested to be. This is especially true as this becomes important later on in the paper during data analysis.

This paragraph has been rewritten as follows to more clearly state the brain regions where we expect to find effects.

The current work tests whether the interaction between memory load and interference during information maintenance differs between young and old adults. We stressed the available neural resources during a verbal DIR task by manipulating the number of items to remember and by introducing an interference numerical addition task during the information delay period. The hypothesis is that older adults employ additional neural resources within the prefrontal cortex and medial temporal lobe to combat increases in age-related intrinsic noise during performance of a working memory task. Engagement of these resources would be comparable to the brain regions that young adults employ when faced with explicit interference with information maintenance (Ma et al., 2014). 

• Line 88-91 – Were participants matched for things like intelligence, education, etc? If so, authors should describe this matching. If not, they should explain why matching was not necessary for this investigation.

 No matching took place and the following text is added to the manuscript.

No matching took place between the two age groups, e.g. education, socio-economic status, etc. Future directions with this data will explore how lifetime exposures impact brain and cognitive measures in older adults. Therefore, matching between age groups was not undertaken.

• Line 103-105 –Authors should described if any of the participants fall below 133 on the Mattis Dementia Rating Scale? If so, how many?

This was part of the initial screening of people interested in the study. Therefore, the number of people not eligible to participate because of this criteria is not available. The text in the manuscript has been modified to clarify this.

During the screening global cognitive functioning was also assessed with the Mattis Dementia Rating Scale and a score of at least 133 was required for admission to the study (Mattis, 1988). 

• Line 134 (and the whole design) – How did the authors come to decide on 2 and 6 letters as the 2 levels of memory load? In older studies of working memory, a capacity of 7 was established (Miller, 1956), while more recent work on strictly visual working memory has found the limit is more like 4 (Cowan, 2001). In either case, these load levels are either both under the capacity, or one is above and the other is below capacity, which makes interpretation of these results challenging. Additionally, memory load has been shown on a continuous scale both behaviourally and neurophysiologically (e.g. Vogel & Machizawa, 2004, Figure 3), so it may have been better to use some intermediate levels of load here as well. I am not asking for further data collection here, but the authors should include a discussion of why they made these decisions, as well as how the results may have differed based on different decisions having been made.

This is a great point. We have now added the following paragraph to the discussion section.

The current work used two levels of memory load: 2 and 6 letters. Previous work with a similar task has used 5 or 6 letters as the maximal load level (Sakai et al. 2002; Zarahn et al. 2006). At this memory load both young and old adults had some level of decrease in accuracy. The intent of the current work was to implement a task that was demanding; however, not too demanding as to exceed cognitive capacity limits. The addition of a 2 letter memory load condition allows for testing of memory load effects. The use of 2 letters was considered low enough to not impact accuracy; however, high enough to keep participants engaged with the task. An ideal experimental design would use additional memory load levels to further explore how behavior and brain activity dynamically change as cognitive demands increase (Schneider-Garces et al. 2010; Vogel and Machizawa 2004). 

• Line 150-159 – The authors should clarify the details of the ‘math’ in this task. Upon first reading, it seemed that answers and digits were chosen randomly from distributions (as described), which would have meant that almost all of the examples given to participants would have been ‘incorrect’. Having looked at this again, I no longer think this was true, but I also am not sure about this. The authors should clearly set out the proportion of trials that were ‘correct’ and ‘incorrect’, and then describe the specific way in which the numbers were acquired from said distributions.

This section was clarified as follows:

Careful consideration was made when selecting the numbers to add and the answers. The summation of five single digits (excluding zero) drawn from a uniform distribution is a Gaussian distribution with a mean of 25. This makes extreme value answers, below 10 or above 40, highly unlikely. In order to minimize this bias and the predictability of the answers, the answers were chosen from a uniformly distributed set of numbers with a range of 5 to 45. These are the minimum and maximum values possible when adding five single digit numbers. Five randomly selected single digits that had this value as a sum were used as the number list. The proportion of positive and negative probe trials was matched at fifty percent. 

• Throughout the results, it would be good to see an estimate of effect size along with the other statistical information that is provided.

We have now added effect sizes to the behavioral results. The following text has also been added at the beginning of the results section to explain the effect sizes used.

Details for these results are reported as follows with unstandardized effect sizes. There is currently a lack of consensus for calculating standardized effect sizes based on the manner in which mixed models partition variance (Rights & Sterba, 2019). Therefore, current recommendations are followed (Pek & Flora, 2018) and unstandardized effect sizes are reported in their original units of seconds or percent accuracy. 

• It is also now very close to standard that data (and analysis code, where available) are uploaded to a repository such as OSF at the time of review. I appreciate that the corresponding author is willing to share the data if requested, but increasing transparency further by uploading it publicly would be better.

The data used in this manuscript is now publicly available at the Open Science Framework. The link is: https://osf.io/xwajn/

The repository contains all participant level contrast maps, behavioral data and a link to the software used for administering this experiment.

• Line 249-250 – The description of the age group x memory load interaction seems to be missing its p value.

 This has been fixed. Thank you for catching this.

• Line 376-378 – If the older adults are showing signs of speed/accuracy trade-off, it would be good for the authors to examine this by looking at something like Inverse Efficiency Scores for both younger and older adults.

This is an excellent suggestion, thank you. Inverse efficiency scores were calculated and analyzed. The following section is now added to the results section.

To explore the finding of better, but slower, responses on the interference task by the old adults the inverse efficiency score was calculated by dividing mean accuracy by median response times (Hughes et al. 2014). This is a method of testing for speed-accuracy tradeoffs. A two-by-two repeated measures ANOVA demonstrated a significant three-way interaction between age group, interference level and memory load (F(1, 37) = 4.47, p = 0.041, η2p=0.11). There was also a main effect of interference level (F(1, 37) = 466.2, p < 0.001, η2p = 0.93). All two way interactions were non-significant; memory load and interference level (F(1, 37) = 4.08, p = 0.051, η2p = 0.10), age group and interference level (F(1, 37) = 0.74, p = 0.40, η2p = 0.02) age group and memory load (F(1, 37) = 1.99, p = 0.17, η2p = 0.051). The main effect of memory load was non-significant (F(1,37) = 3.22, p = 0.07, η2p = 0.08). Marginal means for the three way interaction demonstrate that at high levels of interference, when memory loads increase the IES of the older adults significantly declines (mean difference = -0.35, t = 3.54, pBonf = 0.02), while in the young adults it does not significantly change (mean difference = -0.014, t = 0.15, pBonf = 1.00). 

The following sentence is now added to the discussion. 

Analysis of the inverse efficiency score supports the hypothesis that the old adults are making a speed accuracy tradeoff when completing the interference task.

• Line 386-390 – This is where having used more/intermediate levels of memory load in the experimental design would have been helpful in having more diverse results to examine.

I completely agree. In my current study I am using verbal and spatial delayed match to sample tasks where I am using five levels of task demand. I am also titrating demand to each individual’s cognitive capacity and then scanning them. This study is underway and unfortunately on hold due to COVID-19. Stay tuned!

• Line 391-392 – The phrasing in this sentence is strange to me. I believe I know what the authors intended to say, but to say that ‘increasing memory load and interference share neural resources’ is a bit opaque. Maybe better to say that memory and interference share resources first, and then describe increases/decreases in each

 These sentences have been clarified as follows:

This interpretation suggests that increases in memory load and increases in interference both place greater demands on the neural resources within the same regions. This is supported by the fact that the increases in brain activity from the main effect of interference are all within regions also identified as having greater brain activity with the main effect of memory load. The only exception is a cluster in the superior temporal gyrus. 

• Line 411-412 – This statement could be expanded on to further explain what the general findings are in terms of dual-task effects.

 This statement, and the one previous to it, have been modified as follows:

With this in mind, the current brain imaging results are in line with other findings from dual-task paradigms demonstrating increased activation within inferior frontal and parietal cortices (Szameitat et al., 2002). Age effects in dual-task studies appear to be limited such that the neural substrates utilized when managing dual-tasks are observed as largely similar across age groups (Hartley et al. 2011; Van Impe et al. 2011).

• Line 425-426 – This is an important thing to consider, and while I appreciate the reasons for not analysing things at each stage of the design, I think it may be a useful analysis to include if it is possible to do at this point. In a related study that we did (Wilbiks & Dyson, 2016), we found that individual differences at different phases of the perception/memory process were predictive of capacity for audiovisual integration. A similar finding would be really interesting here!

I agree that teasing apart the different phases of the recognition memory process is very interesting. Unfortunately the hemodynamic response from fMRI is extremely slow. The result is that the measures of brain activity from different task phases all blend together. This becomes a challenge during statistical modeling. Some authors have added extra regressors to models whose only purpose is to “soak” up the variance that bleeds between the regressors during modeling (Zarahn et al., 2006). Bart Rypma also designed experiments with partial trials. These are trials where some trials only have a stimulus phase, some only a stimulus and retention phase and some have no retention phase (Bennett et al., 2013; Motes & Rypma, 2009). I implemented this experimental protocol and piloted it. The code is here:

https://github.com/steffejr/ExperimentalStimuli/tree/master/PartialTrialDIR. After testing five participants, I discovered that the task was confusing for participants as far as what they were to do. The trials with no probe made participants think that something was wrong. This was even if we trained them and explained the experiment in detail. In addition, even though there was less collinearity in this design, it was not removed. I decided that the burden of such a design was not worth the small decrease in collinearity. 

If I were to accept the problems from the collinearity between regressors I would need to reanalyze all the participant level data, which is quite extensive. However, the greater burden is trying to interpret and collate the results. The analyses would either be three different 2x2x2 ANOVAs, one for each task phase, or one 2x2x2x3 ANOVA. I would then be burdened by even less statistical power and the worry that the collinearity is producing spurious findings. 

I therefore feel that exploring the finer grained dynamics of memory recognition and the effects of interference is better left to researchers using EEG and/or MEG.

Review Signed: Jonathan Wilbiks, 8 June 2020.

Reviewer #2: The present study compared differences between younger and older adults in performance of verbal working memory with interference tasks during the delay period. The authors found that the two groups had significant behavioral difference but not much difference in brain activation. They concluded that such null effects in brain activation were due to compensation mechanisms for the older adults. Although the results are generally reasonable, the interpretations and the statistical methods need revision.

1) The results here were not adequately presented. For example, the behavioral results did not indicate those significant effects. What’s the p value? What’s the effect size?

This has been fixed. Thank you for pointing out our omission.

We have now added effect sizes to the behavioral results. The following text has also been added at the beginning of the results section to explain the effect sizes used.

There is a lack of consensus for calculating standardized effect size based on the manner in which mixed models partition variance(Rights & Sterba, 2019). Therefore, current recommendations are followed (Pek & Flora, 2018) and unstandardized effect sizes are reported in their original units of seconds or percent accuracy. 

2) Imaging results were not presented with a standard form. For example,”Brain imaging results were first interpreted using a correction for multiple comparisons with an activation height threshold of p < 0.001 and cluster extent threshold of p < 0.05. ”

 The rest of that paragraph states:

This extent threshold was achieved using a size limit of k = 153 to minimize false positive findings to an alpha of 0.05. These thresholds were calculated using the updated AFNI tool 3dClusterSim and 10,000 Monte Carlo simulations with a non-Gaussian spatial autocorrelation function (ACF) determined based on individual residual maps averaged across all participants with bi-sided thresholding (Robert W. Cox et al., 2017; R. W. Cox et al., 2017). This approach uses updated methods based on findings related to accurate correction for multiple comparisons (Eklund et al., 2016). 

This manner of thresholding neuroimaging data is the current recommended method. As discussed in the highly influential paper by Eklund (2016) there is a major issue in the field with false-positive rates. Robert Cox and the AFNI team responded by implementing appropriate procedures to address these concerns. The methods chosen for the current work adopt these procedures and recommendations.

3) References were carelessly inserted. For example, the first reference they used was “Author correction”instead of normal article. 

 This oversight has been addressed, thank you for pointing it out. 

“These changes include decreased functional neural efficiency and capacity within brain regions required for a task, [3] or declining”[3] was in the wrong place.

The comma position has been modified. The rest of the citations have also been carefully checked. 

4) The interpretation of the results could be alternative. The tasks are simple especially for the load 2 condition. There could be floor effect for the two groups. Imaging results are further less sensitive than the behavioral results, so as not to differentiate between the two groups.

The use of a simple condition is an important design feature of this experiment. It serves as an active baseline comparison for the sixe letter memory load condition. Without the low level condition results are confounded by common sensorimotor and cognitive processes. These include seeing the stimulus letters on the screen and preparing a motor response. The higher level of task difficulty, 6 letters, significantly impacts task performance and has shown significant age effects in previous work where an interference condition was not used (Zarahn et al., 2006).

Differential sensitivity between the neural and behavioral measures is a real possibility. This is the reason why the experiment was so carefully designed based off of one million simulations as discussed in the “Model Efficiency” section. Furthermore, this is also why the statistical thresholds for the neuroimaging data were carefully chosen using current recommendations by the field and 10,000 Monte Carlo simulations based off of our neuroimaging data’s residual maps.

Based on these careful considerations with experimental design and analysis we are confident in our interpretation of the results. In addition, based on comments from Reviewer #1, further discussion is now made about how our findings are similar to other studies where age related behavioral but not neuroimaging results have been found. 

5) The degree-of-freedom in the statistical results is weird for like thousands.

The current analyses use mixed effects modeling. With mixed effects modeling the data is entered in long format; therefore, every trial is a row of data. Therefore, when degrees of freedom are calculated they are based first off of the total number of trials of data. This number is adjusted by the number of factors in the model and by other adjustments derived from the standard deviation of the data using the Satterthwaite method. This degrees of freedom calculation method has been stated in the manuscript along with a citation.

6. PLOS authors have the option to publish the peer review history of their article (what does this mean?). If published, this will include your full peer review and any attached files.

Do you want your identity to be public for this peer review? For information about this choice, including consent withdrawal, please see our Privacy Policy.

Reviewer #1: Yes: Jonathan M. P. Wilbiks

Reviewer #2: No

Bennett, I. J., Rivera, H. G., & Rypma, B. (2013). Isolating age-group differences in working memory load-related neural activity: assessing the contribution of working memory capacity using a partial-trial fMRI method. NeuroImage, 72, 20–32.

Cox, R. W., Chen, G., Glen, D. R., Reynolds, R. C., & Taylor, P. A. (2017). FMRI Clustering in AFNI: False-Positive Rates Redux. Brain Connectivity, 7(3), 152–171.

Cox, R. W., Chen, G., Glen, D. R., Reynolds, R. C., & Taylor, P. A. (2017). fMRI clustering and false-positive rates [Review of fMRI clustering and false-positive rates]. Proceedings of the National Academy of Sciences of the United States of America, 114(17), E3370–E3371.

Eklund, A., Nichols, T. E., & Knutsson, H. (2016). Cluster failure: Why fMRI inferences for spatial extent have inflated false-positive rates. Proceedings of the National Academy of Sciences of the United States of America. https://doi.org/10.1073/pnas.1602413113

Ma, W. J., Husain, M., & Bays, P. M. (2014). Changing concepts of working memory. Nature Neuroscience, 17(3), 347–356.

Motes, M. A., & Rypma, B. (2009). Working memory component processes: isolating BOLD signal changes. NeuroImage, 49(2), 1933–1941.

Pek, J., & Flora, D. B. (2018). Reporting effect sizes in original psychological research: A discussion and tutorial. Psychological Methods, 23(2), 208–225.

Rights, J. D., & Sterba, S. K. (2019). New Recommendations on the Use of R-Squared Differences in Multilevel Model Comparisons. In Multivariate Behavioral Research (pp. 1–32). https://doi.org/10.1080/00273171.2019.1660605

Zarahn, E., Rakitin, B., Abela, D., Flynn, J., & Stern, Y. (2006). Age-related changes in brain activation during a delayed item recognition task. Neurobiology of Aging, 28(5), 784–798.

---

## [Decision Letter · Decision Letter 1]

16 Jul 2020

Neural capacity limits on the responses to memory interference during working memory in young and old adults

PONE-D-20-11960R1

Dear Dr. Steffener,

We’re pleased to inform you that your manuscript has been judged scientifically suitable for publication and will be formally accepted for publication once it meets all outstanding technical requirements.

Kind regards,

Stephen D. Ginsberg, Ph.D.

Section Editor

PLOS ONE

**Comments to the Author**

1. If the authors have adequately addressed your comments raised in a previous round of review and you feel that this manuscript is now acceptable for publication, you may indicate that here to bypass the “Comments to the Author” section, enter your conflict of interest statement in the “Confidential to Editor” section, and submit your "Accept" recommendation.

Reviewer #1: (No Response)

Reviewer #2: All comments have been addressed

2. Is the manuscript technically sound, and do the data support the conclusions?

Reviewer #1: Yes

Reviewer #2: Partly

3. Has the statistical analysis been performed appropriately and rigorously? 

Reviewer #1: Yes

Reviewer #2: N/A

4. Have the authors made all data underlying the findings in their manuscript fully available?

Reviewer #1: Yes

Reviewer #2: Yes

5. Is the manuscript presented in an intelligible fashion and written in standard English?

Reviewer #1: Yes

Reviewer #2: Yes

6. Review Comments to the Author

Reviewer #1: Review: Neural capacity limits on the responses to memory interference during working memory in young and old adults. (PONE-D-20-11960R1)

The authors have improved this manuscript by completing a thorough revision. I am now happy to support publication of this manuscript after one very minor revision. While I thank the authors for providing their data and code on OSF, I think it would be important to include (perhaps at the beginning of the results section?) the URL so that readers can easily access the data.

I will look forward to seeing your ongoing project using five levels of task demand on verbal and spatial tasks – sounds very interesting, so I hope you can get back to testing sometime soon!

Signed: Jonathan Wilbiks, 29 June 2020

Reviewer #2: (No Response)

7. PLOS authors have the option to publish the peer review history of their article (what does this mean?). If published, this will include your full peer review and any attached files.

Reviewer #1: **Yes: **Jonathan Wilbiks

Reviewer #2: No

---

## [Editor Report · Acceptance letter]

22 Jul 2020

PONE-D-20-11960R1 

Neural capacity limits on the responses to memory interference during working memory in young and old adults 

Dear Dr. Steffener:

I'm pleased to inform you that your manuscript has been deemed suitable for publication in PLOS ONE. Congratulations! Your manuscript is now with our production department. 

Kind regards, 

on behalf of

Dr. Stephen D. Ginsberg 

Section Editor

PLOS ONE